

**Impact   of volcanic sulfate aerosols on the stratospheric heating: implications on the**
**Quasi-Biennial Oscillation**
Prashant Chavan[1,2], Suvarna Fadnavis[1*], Anton Laakso[3], Jean-Paul Vernier[4,5],
Simone Tilmes[6], and Rolf Müller[7]
[1] Indian Institute of Tropical Meteorology, Center for Climate Change, MoES, India
[2]Department of Atmospheric and Space Sciences, Savitribai Phule Pune University, Pune,
India
[3]Finnish Meteorological Institute, Kuopio, Finland
[4] GSMA, UMR CNRS 7331, Université de Reims Champagne-Ardenne, Reims, France
[5]NASA Langley Research Center, Hampton, VA, USA
[6] National Center for Atmospheric Research, Boulder, CO, USA
[7] Forschungszentrum Jülich GmbH, ICE-4, Jülich, Germany
*corresponding author email: suvarna@tropmet.res.in

**Abstract:**
Large and moderate volcanic eruptions significantly impact Earth's atmosphere by releasing
sulphur emissions, thereby affecting atmospheric dynamics and QBO. Using the ECHAM6-
HAMMOZ model, we show the impact of eruptive volcanoes on the tropical stratosphere and
Quasi-biennial oscillation (QBO) from 2001 to 2013. Our simulations with volcanoes, when
compared without volcanoes, show that volcanic sulfate aerosols enhance the stratospheric
aerosol optical depth (SAOD) two months after the eruption of Rabaul (0.0034); Sarychev
(0.0040) and Nabro (0.0097). The enhanced SOAD in the tropics (0.0014) led to a radiative
forcing at the top of the atmosphere (TOA) by -0.92±0.34 W m$^{-2}$ and at the surface by -
0.88±0.18 W m$^{-2}$ in the tropical region. The volcanic aerosol precursors enter the tropical
stratosphere, propagating upward and enhancing sulfate aerosol concentrations by 46.95 ng
m$^{-3}$ and heating rates by 0.13±0.05×10$^{-2}$ K d$^{-1}$.  The QBO estimated from model simulations
using the wavelet analysis shows that stratospheric heating caused by the volcanoes reduces
the amplitude of the QBO and disrupts its phases, resulting in the prolongation of the easterly



phase by ~12 to 20 months and the westerly phase by ~16 to 24 months. The secondary
meridional circulation induced by the QBO produces the double-peak structure in the
amplitude near the equator, with peaks at 10 hPa and at 50 hPa. Our study points out that
moderate and large volcanoes modulate the QBO. Since QBO also modulates tropical
convection and weather, we suggest including volcanic eruptions and the QBO in weather
prediction models for a better forecast.

Keywords: Volcanoes, Quasi-biennial oscillation (QBO), stratospheric heating, sulfate
aerosols.

**1. Introduction**
Strong and moderate volcanic eruptions inject sulphur dioxide ($SO_2$) gas into the
stratosphere, which produces sulfate aerosol particles that contribute to stratospheric heating
by absorption of infrared radiation, thus influencing stratospheric dynamics (Hegerl et al.,
2003; Sigl et al., 2015; Santer et al., 2015). The sulfate aerosols originating from $SO_2$ injected
by the volcanoes into the stratosphere reside for about 1–5 years (Carn et al., 2016). The sulfate
aerosols scatter (shortwave) sunlight, resulting in a net negative radiative forcing at the top of
the atmosphere (TOA) and a cooling of the surface (Kremser et al., 2016).

Over the past decades, multiple moderate volcanic eruptions (VEI≥3) have augmented
the stratospheric aerosol load, resulting in the enhancement of SAOD (Vernier et al., 2011) and
cooling of the surface (Kremser et al., 2016). A number of satellite observations shows an
enhancement in SAOD; for example, CALIPSO satellite data showed an increase in SAOD
(0.006 to 0.012) following multiple volcanic eruptions during 2008 – 2012 (Andersson et al.
2015). CALIPSO satellite observations show an increase in SAOD ~40 % (0.008) compared



to the background level during 2013 (Friberg et al. 2018). Similarly, combined satellite data
from SAGE–II, GOMOS, and CALIPSO showed an increase in SAOD of $20.4 \times 10^{-5}$ yr$^{-1}$ from
2000 – 2009, primarily driven by a sequence of medium-size volcanic eruptions (Vernier et al.,
2011). The model simulation studies also reported a similar increase in global and tropical
mean SAOD by volcanic eruptions (see Table 1).

Table 1: SAOD from model simulations for a series of volcanoes

| Model | Period | SAOD Increase | Region | Reference |
|---|---|---|---|---|
| CESM1 and WACCM–MAM | 2005 – 2014 | 0.0076 | Global | Schmidt et al., 2018 |
| WACCM–CARMA | 2013 – 2019 | 0.003 to 0.02 | Global | Tidiga et al., 2022 |
| CESM1(WACCM) | 1990 – 2014 | 0.01 | Global | Mills et al., 2016 |
| CMIP6 | 2005 – 2014 | 0.007 | Global | Schmidt et al., 2018 |
| GEOS–Chem | 2005 – 2012 | 0.001 to 0.01 | Global | Ge et al., 2016 |
| HadGEM2–ES | 2000 – 2013 | 0.009 | Tropical | Haywood et al., 2014 |
| EMAC | 1990 – 2019 | 0.005 to 0.4 | Tropical | Schallock et al., 2023 |
| EMAC | 2002 – 2012 | 0.001 – 0.01 | Tropical | Brühl et al., 2015 |

The enhancement in SAOD by volcanoes resulted in a reduction in radiative forcing at the TOA
by 0.1 to 0.2 W m$^{-2}$ during 1990 – 2019. For example, Schmidt et al. (2018) estimated a
reduction in global mean radiative forcing by 0.10 W m$^{-2}$ during 2005 – 2015 using the
CESM1-WACCM model by comparing simulations with and without volcanic sulfur dioxide
emissions. Similarly, Schallock et al. (2023) estimated a global instantaneous radiative forcing
of -0.1 Wm$^{-2}$ during 1990 – 2019 from minor eruptions, relative to a background stratospheric



aerosol forcing of about -0.04 Wm⁻², using the EMAC model simulation. Ridley et al. (2014)
estimated RF -0.19 W m⁻² from 2000 – 2013 using the Earth Model of Intermediate Complexity
(EMIC). A number of satellite observations show a reduction in global mean radiative forcing
following volcanic eruptions. For example, studies by Friberg et al. (2018) and Andersson et
al. (2015) reported using CALIPSO satellite data, which shows decreases in radiative forcing
after volcanic eruptions: 0.45 W m⁻² after the combined eruptions of Okmok (July 2008) and
Kasatochi (August 2008);-0.5 W m⁻² after Sarychev (June 2009); 0.25 W m⁻² after Merapi
(November 2010); and 0.35 W m⁻² after both Puyehue-Cordón Caulle and Nabro (June 2011),
with all values representing the monthly mean one month after each eruption.

The sulfate aerosols absorb solar near-infrared radiation and cause heating in the layer

where they reside (Kremser et al., 2016; Fadnavis et al., 2021a). Past studies show that tropical
volcanoes enhanced heating rates considerably in the tropical stratosphere e.g. Pinatubo (1991)
enhanced heating rates in the tropics by 0.28 K d⁻¹, Ruiz (1986) by 0.023 K d⁻¹, El Chichón
(1982) by 0.0045 K d⁻¹, and Agung (1963) by 0.075 K d⁻¹, Nabro (2011) by 0.003 K d⁻¹ (Pitari
et al. 2016, Fairlie et al. 2014, Fadnavis et al 2021a). Additionally, Schallock et al. (2023)
reported that moderate volcanic eruptions during 1990 - 2019 enhanced heating rates in the
tropics by 0.03 K d⁻¹. The stratospheric heating caused by extratropical volcanic aerosols is
reported by Toohey et al. (2014). Occasionally, these aerosols are transported to the tropics
(Brühl et al., 2015; Oman et al., 2006). Thus, these aerosols produce heating in the tropics
(Robock, 2000; Timmreck, 2012).

The volcanic sulfate aerosol-induced heating in the stratosphere disrupts the thermal

structure (Santer et al., 2015) and influences the QBO (Bittner et al., 2016; Brenna et al., 2019;
Toohey et al., 2014; Diallo et al., 2018). The impact of large volcanoes on the QBO is reported



in a few studies, e.g., Labitzke, 1994 reported that Mount Pinatubo in 1991 induced warming
of 3 K in the stratosphere that led to prolonged easterly phase of the QBO. The ECHAM5
model simulations showed a shift in QBO phases from easterly to westerly after the Mount
Pinatubo eruption (Thomas et al., 2009). Brenna et al. (2021) reported that the Los Chocoyos
eruption (VEI=8) (14.6°N, 91.2°W) disrupted the QBO periodicity for up to 10 years. Apart
from volcanic aerosols, geoengineering studies wherein injection of 2.5 to 5 Tg of $SO_2$ into the
stratosphere have also shown impacts on the QBO phase induced by stratospheric heating
(Aquila et al., 2014; Pitari et al., 2016). Richter et al. (2017) found that in the CESM1
(WACCM) simulation with an equatorial injection of 12 Tg $SO_2$ per year, changes to the QBO
phase resulted in an increased period of approximately 3.5 years. Several studies have reported
a prolonged westerly phase of the QBO (Aquila et al., 2014). In extreme cases, some studies
even show a complete shutdown of the QBO. For example, simulations using the ECHAM
model indicate that a large injection of 8 Tg of $SO_2$ can cause this shutdown, while the
WACCM model shows the QBO shutting down with a smaller injection of just 2 Tg of $SO_2$
(Niemeier and Schmidt, 2017; Niemeier et al., 2020). The shutdown of the QBO not only
depends on the injection rate and the model used for simulation but also on the injection
strategy, including factors such as location, altitude, and frequency of injections (Laakso et al.,

2022).


Here, we aim to investigate the influence of moderate volcanoes during 2001 – 2013 on

the QBO using the state-of-the-art ECHAM6-HAMMOZ chemistry–climate model. During
this period, there were a total of 20 volcanic eruptions (VEI≥3). Among these, 10 were in the
tropics (20º S – 20º N), 8 in the northern extra-tropics (20º N – 90º N), and 2 in the southern
extra-tropics (20º S – 90º S) (see Figure 1). The structure of this paper encompasses various
sections: Section 2, the model setup, and details of the simulations performed for this





investigation. Subsequently, Section 3 presents and discusses the outcomes derived from these
simulations. Finally, in Section 4, the conclusions are made.

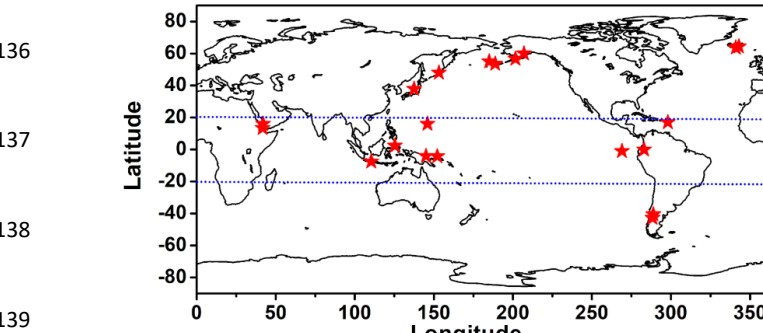

Fig. 1: Spatial distribution of the moderate and large (VEI ≥ 3) volcanic eruptions. Red stars
indicate the location of volcanoes. Blue dotted line indicates the tropical region (20° S – 20°
N).

**2. Model simulations and data analysis:**
**2.1 Model description and experimental set-up**
We use the state-of-the-art aerosol–chemistry-climate model ECHAM6–HAMMOZ. It
comprises the general circulation module ECHAM6, coupled to the aerosol and cloud
microphysics module HAM (Stier et al., 2005; Tegen et al., 2019). HAM predicts the
nucleation, growth, evolution, and sinks of sulfate, black carbon (BC), particulate organic
matter (POM), sea salt (SS), and mineral dust (DU) aerosols. The size distribution of the
aerosol population is described by seven log-normal modes with prescribed variance as in the
M7 aerosol module (Stier et al., 2005; Zhang et al., 2012). Moreover, HAM explicitly simulates
the impact of aerosol species on cloud droplets and ice crystal formation. Other details of the
model and emissions are reported by Fadnavis et al., 2017, 2019, 2021b 2021c. Anthropogenic
aerosol emissions are from a Community Emissions Data System (CEDS). The volcanic SO$_2$





emissions inventory used in model simulations as per Carn et al. (2017). The total annual
amount and the explosive annual amount of global volcanic SO$_2$ emissions used in the model
are tabulated in Table 2.
Table – 2. Volcanic SO$_2$ (kt) emissions reaching the stratosphere and the uppermost tropical
troposphere from the Volcanic Sulfur Emission Inventory. The total annual amount and the
explosive annual amount of global volcanic SO$_2$ emissions were calculated from satellite
observations from 1979 to 2014 by Carn et al. (2016).

| Year | Carn et sl., (2016): total | Carn et al., (2016): explosive |
|---|---|---|
| 2000 | 653 | 326 |
| 2001 | 1783 | 122 |
| 2002 | 2626 | 271 |
| 2003 | 679 | 271 |
| 2004 | 2997 | 410 |
| 2005 | 4634 | 2501 |
| 2006 | 1347 | 661 |
| 2007 | 712 | 122 |
| 2008 | 2624 | 2318 |
| 2009 | 1934 | 1379 |
| 2010 | 1470 | 867 |
| 2011 | 6030 | 4310 |
| 2012 | 763 | 563 |
| 2013 | 185 | 563 |

The model simulations are performed at the T63 spectral resolution corresponding to
1.875°x1.875° in the horizontal dimension, while the vertical resolution is described by 47
hybrid σ−p levels from the surface up to 0.01 hPa (approx. 80 km). The simulations have been
carried out with a time step of 20 min. Monthly varying Atmospheric Model Intercomparison
Project (AMIP) sea surface temperature (SST) and sea ice cover (SIC) (Taylor et al., 2000)
were used as lower boundary conditions. We performed two experiments aerocom UTLS
(https://wiki.met.no/_media/aerocom/A3_UTLS_2019-11-26.pdf): (1) a VAL simulation
where all aerosols, including volcanic emissions, are included and another experiment where
(2) all volcanic aerosol emissions are switched off all over the globe during the study period
(2001 – 2013) (referred to as VAL0). The simulation was performed from 1 January 2001 to



December 2013 from stabilized initial fields created after model integration for one year. Dust
emission parameterization is the same in all the simulations and is based on earlier work (Tegen
et al., 2002). In volcano simulations, we assumed that all volcanoes erupted at respective times
and locations. There are 20 volcanoes globally (see Figure 1) in the study period, including
large and moderate volcanoes (VEI≥3) (see Table 3).
Table – 3: List of volcanoes from 2001 to 2013 and their Volcanic Explosivity Index (VEI)

| Sr. No. | Volcano | Month Year | Location | VEI |
|---|---|---|---|---|
| 1 | Shiveluch (Sh) | June 2001 | 56.65 °N, 161.36° E | 4 |
| 2 | Ruang (Ru) | September 2002 | 2.30 °N, 125.37°E | 4 |
| 3 | Reventador (Rv) | November 2002 | 0.08 °S, 77.66 °W | 4 |
| 4 | Anatahan (An) | May 2003 | 16°N, 146°E | 3 |
| 5 | Manam (Ma) | November 2004 | 4.08 °S, 145.04 °E | 4 |
| 6 | Sierra Negra (Si) | Oct 2005 | 1°S 91°W | 3 |
| 7 | Soufrière Hills (So) | May 2006 | 17°N, 62°W | 3 |
| 8 | Rabaul (Ra) | October 2006 | 4.27 °S, 152.20 °E | 4 |
| 9 | Jebel at Tair (Je) | Sep 2007 | 16°N, 42°E | 3 |
| 10 | Chaiten (Ch) | May 2008 | 42.83 °S, 72.65 °W | 4 |
| 11 | Mt okmok (Ok) | July 2008 | 53.48 °N, 168.17 °W | 4 |
| 12 | Kasatochi (Ka) | August 2008 | 55.00 °N, 175.00 °W | 4 |
| 13 | Redoubt (Re) | Mar 2009 | 60 °N, 153°W | 3 |
| 14 | Sarychev (Sa) | June 2009 | 48.00 °N, 153.20 °E | 4 |
| 15 | Eyjafjallajokull (Ey) | April 2010 | 63.63 °N, 19.60 °W | 4 |
| 16 | Merapi (Me) | November 2010 | 7.54 °S, 110.44 °E | 4 |
| 17 | Grimsvotn (Gr) | May 2011 | 64.42 °N, 17.33 °W | 4 |
| 18 | Nabro (Na) | June 2011 | 13.37 °N, 41.70 °E | 4 |
| 19 | Puyehue-Cordon Caulle (Pu) | June 2011 | 40.59 °S,72.12 °W | 5 |
| 20 | Etna (Et) | April 2013 | 37.89 °N, 137.48 °E | 3 |

**2.2 The Global Space-based Stratospheric Aerosol Climatology (GloSSAC)**
We compared simulated SAOD with Global Space-based Stratospheric Aerosol
Climatology (GloSSAC) SAOD. The GloSSAC provides a 43–year record of stratospheric
aerosol properties, focusing on aerosol extinction coefficients at 525 nm and 1020 nm. This





climatology is primarily based on data from the Stratospheric Aerosol and Gas Experiment
(SAGE) instruments until mid-2005, followed by the Optical Spectrograph and InfraRed
Imager System (OSIRIS) and the Cloud-Aerosol Lidar and Infrared Pathfinder Satellite
Observation (CALIPSO) from mid-2017 onwards. GloSSAC also integrates data from other
satellite platforms, as well as ground-based, airborne, and balloon-borne instruments, to
achieve a global and temporally continuous dataset. The dataset spans from 1979 to 2023,
covering latitudes from 80°S – 80°N, with a horizontal resolution of 500 to 1000 km and a
vertical resolution from 100 meters to less than 1 km, offering a comprehensive representation
of stratospheric aerosol variability (Kovilakam et al., 2023; Thomason and Knepp, 2023;
NASA/LARC/SD/ASDC, 2022). Here, we have used the data of GloSSAC Version 2.2 SAOD
at 525 nm from the tropopause to 40 km for the period 2001 – 2013
(https://asdc.larc.nasa.gov/project/GloSSAC).
**2.3 ERA5 Reanalysis data**
We compared simulated zonal winds with ERA5 5th generation of the European Centre for
Medium-Range Weather Forecasts (ECMWF). Monthly mean data were obtained at horizontal
grids of 0.25° × 0.25° and the heights covering from 1000 hPa to 1 hPa at 37 pressure levels
(Hersbach et al., 2020). We analyzed monthly mean zonal winds data at 200 hPa and 10 hPa
for the year 2008 (https://cds.climate.copernicus.eu/datasets/reanalysis-era5-pressure-levels-
monthly-means?tab=download).
**2.4 Wavelet analysis**
We analyse the anomalies obtained as VAL – VAL0 simulations to understand the impact
of volcanic sulfate aerosols. QBO is not included in the ECHAM6-HAMMOZ model used in
this study; wavelet analysis has been used to detect the amplitude and phase of the QBO from
the simulated zonal wind from January 2001 to December 2013. The application of wavelet

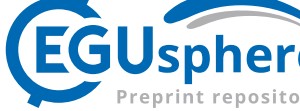

transform on the simulated winds is as per Torrence and Compo (1998). We used the Morlet
wavelet as the basic wavelet, which consists of a plane wave modulated by a Gaussian.
$$\Psi_0 = \pi^{\frac{1}{4}} e^{iw_o \eta} e^{\frac{-n^2}{2}} \qquad (1)$$

In equation (1) $\omega_0$ is the non-dimensional frequency. We have taken $\omega_0 = 6$ and $\eta$ is a non-
dimensional time parameter. The continuous wavelet transform of a discrete time series $X_n$, is
defined in equation (2)
$$W_n(s) = \sum_{n'=0}^{n-1} X_n \Psi^* \left[ \frac{(n'-n)\delta}{t} s \right] \qquad (2)$$

In equation (2) the (*) indicates the complex conjugate and s denotes the wavelet scale. The
amplitude versus scale relationship and the variation of amplitude with time can be obtained
by varying the wavelet scale and translating along the time axis.

In finite-length time series one assumes the data is cyclic hence errors will occur at the
beginning and end of the wavelet power spectrum. To minimize this error padding is done at
the end of the time series by putting zeroes before doing the wavelet transform and then
removing these zeros afterward. The time series is padded with zeros so that the total length of
the time series is up to N, the next-higher power of two. This helps to limit the edge effects and
speeds up the process. The cone of influence (COI) separates the accurate wavelet coefficients
from those that are inaccurate. The edge effects are significant in the region of COI. COI I is
defined here as the e-folding time for the autocorrelation of wavelet power at each scale. This
e-folding time is chosen such that the wavelet power for a discontinuity at the edge drops by a
factor $e^{-2}$ ensuring that beyond this point the edge effects are negligible (Torrence and Compo,
1998). In this study, a comprehensive analysis of the spatiotemporal variations in the amplitude
and phase of the QBO derived from wavelet analysis on simulated zonal winds has been





presented. The positive phase of the QBO in zonal wind represents westerly zonal winds, while
the negative phase corresponds to easterly winds, which are expressed in degrees (0° – 360°).

## 3. Results:

### 3.1 Model evaluation:

Figure 2a compares simulated global mean SAOD (550 nm) with GloSSAC
observations (520 nm) from 2001 – 2013. This figure shows the simulated SAOD
underestimates the GloSSAC SAOD during 2001 – 2013. However, the model SAOD aligns
well with GloSSAC in capturing the peaks and dips qualitatively for the period 2001 – 2013,
which shows good agreement with GloSSAC. Both the model and GloSSAC data show low
SAOD from 2001 – 2005, as it is a volcanically quiescent period, the model slightly
underestimates it by 0.0036 than GloSSAC. From 2006 – 2009, both the model and GloSSAC
indicated a rise in SAOD, with a peak in 2006 attributed to the Rabaul eruption and another in
2009 from the Sarychev eruption.

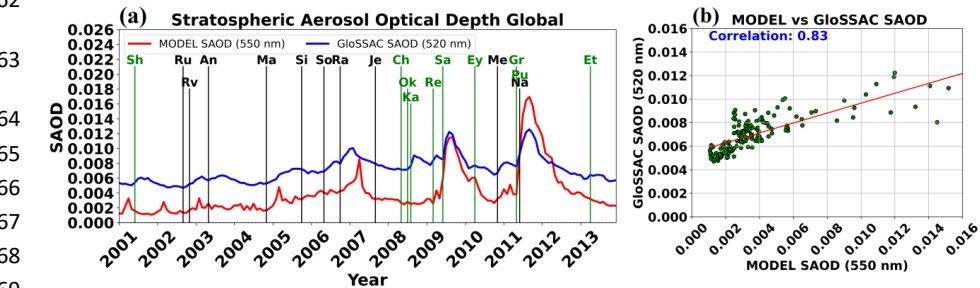

Fig. 2: (a) Time series global mean Stratospheric Aerosol Optical Depth (SAOD) during 2001 -2013, GloSSAC at 525 nm (blue line), ECHAM-HAMMOZ at 550 nm (red line), solid vertical lines show the month of eruption. Black lines indicate the eruption in the tropical region; green lines indicate the eruption in the extratropical region, (b) correlation between SAOD from GloSSAC at 520 nm and ECHAM-HAMMOZ at 550 nm.





The model underestimates SAOD by 0.0015 compared to GloSSAC for the 2006 Rabaul
eruption; however, the model shows good agreement with GloSSAC for the 2009 Sarychev
eruption. A similar SAOD peak is seen for the 2011 Nabro eruption, with the model slightly
overestimating SAOD by 0.0043 compared to GloSSAC. After 2012, both the model and
GloSSAC show a decline in SAOD, although the model consistently underestimates the SAOD
by 0.0021 for the period 2012 – 2013.

Figure 2b shows the correlation between the model and GloSSAC SAOD, with a
correlation coefficient of 0.83. This indicates that the model performs well in comparison to
observations. The underestimation by the model may be due to differences in the model and
GloSSAC; for example, the model outputs SAOD on the pressure levels while GloSSAC gives
them on the altitudes. Also, nitrates and carbonyl sulfide are not included in the ECHAM6–
HAMMOZ model. The model transport processes and resolution can produce the above
observed differences.

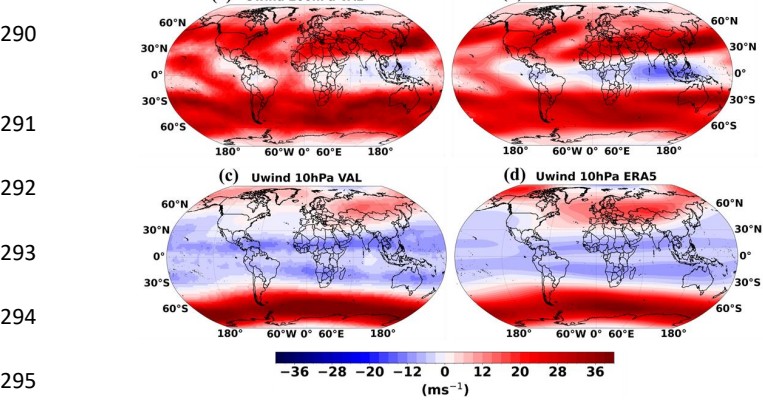

Fig. 3: (a) Spatial distribution of the zonal winds at 200 hPa for the year 2008 from VAL
simulation, (b) same as (a) but from ERA5, (c)-(d) same as (a)-(b) but at 10 hPa.





To evaluate the characteristics of the simulated winds, we compare zonal wind from
ECHAM6–HAMMOZ simulation and ERA5 reanalysis data for the year 2008 at 200 hPa (Fig.
3a-b) and 10 hPa (Fig 3c-d). Figures 3a-d illustrate qualitative agreement between the model
and ERA5 at the respective levels. For example, at 200 hPa, The model and ERA5 show the
strong westerly winds in the mid-latitudes and weaker near the equator. However, the model
underestimates zonal winds in the Northern extra tropical region by 1.5 m s$^{-1}$. While, the model
overestimates over the equatorial region over the Indian Ocean and the western Pacific (by 6.3
m s$^{-1}$), the central and eastern Pacific regions (by 5 m s$^{-1}$) and the Southern extra tropical region
(by 1.05 m s$^{-1}$).

At 10 hPa, patterns of westerly winds in northern and Southern the high latitudes seen in the
model simulations agree with ERA5. Although winds patterns agree qualitatively, the
simulated winds are underestimated with ERA5 in the tropical (by 0.73 m s$^{-1}$) and Northern
extra tropical region (by 2.85 m s$^{-1}$). The model overestimates zonal winds in the Southern
extra tropics by 4.2 m s$^{-1}$. Considering model performance of the model in terms of qualitative
agreement and small under/over estimation we estimate QBO using wavelet analysis from the
simulated zonal winds.

**3.2 Impact of volcanoes on the SAOD, radiative forcing and heating rates**

Figure 4 illustrates simulated anomalies of SAOD (VAL-VAL0) for (1) the global mean

and (2) the zonal mean averaged over the tropical region (20° S – 20° N). Figure 4 depicts that,
during the volcanic quiescent period from 2001 – 2004, there is an increase in SAOD anomalies
in the tropical region by 0.0001 to 0.0008 and globally by 0.0001 to 0.001.  There is an increase



in SAOD in the tropical region by 0.0025 and globally by 0.0035, averaged for 2005 – 2013
due to a series of volcanic eruptions (Table 3).


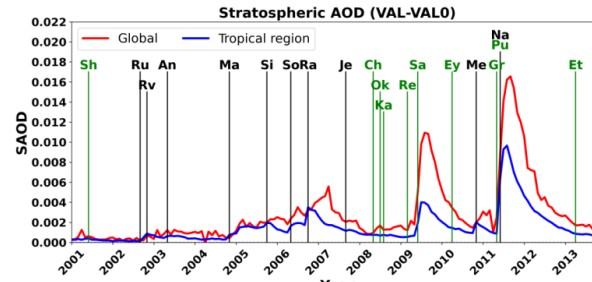







Fig. 4: Time series of Stratospheric Aerosol Optical Depth (SAOD) (above tropopause to 1hPa)
during 2001-2013 using ECHAM6-HAMMOZ model simulations (VAL-VAL0), Global mean
(red line) and zonal mean averaged over the tropical region (blue line) (20° S - 20° N). Solid
vertical lines show the month of eruption. Black lines indicate the eruption in the tropical
region, and green lines indicate the eruption in the extratropical region.


Notably, Figure 4 highlights peaks in SAOD in 2006 due to the Rabaul volcano (0.0034), in
2009 due to the Sarychev volcano (0.0040), and in 2011 due to the Nabro volcano (0.0097) in
the tropical region (Table S1). Andersson et al. (2015) reported that the Kasatochi eruption in
2009, the Sarychev eruption in 2010, and the Nabro eruption in 2011 collectively enhanced
global SAOD by 30% between 2008 and 2012. Our ECHAM-HAMMOZ model simulations
show an increase in SAOD by 38.2% due to Kasatochi, Sarychev, and Nabro eruptions during
2008 – 2012, which is in good agreement with the findings of Anderson et al. (2015), indicating
the reliable performance of the model. Additionally, GOMOS satellite data in the tropical
region show SAOD ranging from 0.002 to 0.0055 during 2002 – 2008 (Vernier et al. 2011),
our model also shows SAOD values ranging from 0.001 to 0.005 during the same period, which
aligns closely with these observations, indicating good agreement.




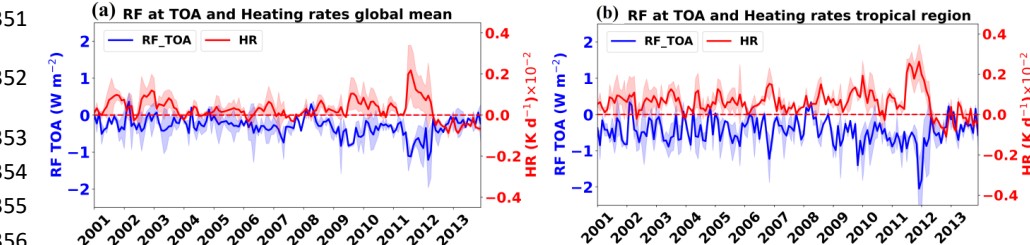

Fig. 5: (a) Global mean radiative forcing at TOA (W m$^{-2}$) (blue line), stratospheric heating rates (K d$^{-1}$) $\times 10^{-2}$ (red line) (above tropopause to 1hPa) from VAL-VAL0 during 2001-2013, (b) same as (a) but zonally averaged over the tropical region (20°S – 20°N).

Further, we show changes in RF and heating rates caused by volcanic eruptions. Figures 5a-b show a decrease in the global mean RF at the TOA by 0.43±0.27 W m$^{-2}$ and by 0.92±0.34 W m$^{-2}$ over the tropics averaged over the years 2001 – 2013. There is a reduction in global mean surface radiative forcing by 0.47±0.25 W m$^{-2}$ and within the tropics by 0.88±0.18 W m$^{-2}$ (Fig. S1). The impacts of tropical volcanoes are prominently visible (Figure 5b) while it is subdued in the global mean (Figure 5a), for example, a drop in RF at the TOA in the tropical region due to Rabaul in 2006 (1.22±0.65 W m$^{-2}$), Sarychev in 2009 (1.4±0.17 W m$^{-2}$), and Nabro in 2011 (2.04±0.81 W m$^{-2}$) (see table S1). The radiative forcing within the tropical region is higher (-0.92±0.34 W m$^{-2}$) than the global mean (-0.43±0.27 W m$^{-2}$) since the majority of volcanoes are tropical and solar intensity is on average, greater in the tropics (Table 3). Past studies also illustrated a similar impact of volcanic eruptions on radiative forcing. For example, the El Chichón eruption in April 1982 caused a reduction in radiative forcing by 2 to 4 W m$^{-2}$ over a year (Robock and Mao, 1995), while the Pinatubo eruption in June 1991 reduced a TOA radiative forcing by 4.5 W m$^{-2}$ within the region 40° S – 40° N (Minnis et al., 1993, Yang et al., 2019).

Figures 5a and 5b also show significant enhancements in stratospheric heating rates (locally) due to volcanic sulfate aerosols. Figure 5a shows an increase in the global mean



heating rates by $0.066\pm0.019\times10^{-2}$ K d$^{-1}$, while within the tropics by $0.13\pm0.05\times10^{-2}$ K d$^{-1}$.
Similar to the distribution of SAOD, a peak in the heating rate in the tropical region due to
tropical volcanoes is more prominent than in the global mean. For example, in 2006, a peak
due to Rabaul ($0.15\pm0.08\times10^{-2}$ K d$^{-1}$), in 2009 due to Sarychev ($0.19\pm0.07\times10^{-2}$ K d$^{-1}$), and
in 2011 due to Nabro ($0.26\pm0.08\times10^{-2}$ K d$^{-1}$) is evident in Figure 5b (see also table S1).
However, the heating rate decreased after 2012 due to the reduced amount of stratospheric
sulfate aerosols (Figure 4b). Due to the higher numbers of tropical volcanoes and SAOD during
the study period, the heating rates within the tropical region ($0.13\pm0.05\times10^{-2}$ K d$^{-1}$) exceed the
global average ($0.066\pm0.019\times10^{-2}$ K d$^{-1}$). Previous studies, such as Fadnavis et al. (2021a),
demonstrated enhancement of heating rates due to Nabro volcanic aerosols during the July-
September 2011 by 0.01 K d$^{-1}$ at an altitude of 15 – 20 Km and 0.003–0.005 K d$^{-1}$ at 20 – 35
Km which are in agreement with the current study.

In this section, we analyze the distribution of vertical velocity and volcanic sulfate
aerosols in the tropical stratosphere. Figure 6a depicts the time series of changes in vertical
velocity (VAL - VAL0) averaged over the tropics (20° S – 20° N) for the period 2001 – 2013.
Figure 6 shows that vertical velocities substantially increase at the altitudes above the
tropopause by 0.42 m s$^{-1}$. The heating in the lower stratosphere plays a crucial role in
facilitating upward transport (Brühl et al., 2015; Toohey et al., 2014). Figure 6 b illustrates the
large enhancement of sulfate aerosols in the stratosphere following volcanic eruptions. This
figure shows less enhancement in sulfate aerosols (27.62 ng m$^{-3}$) in the volcanically quiescent
period from 2001 – 2005 in the VAL-VAL0 simulation.However, there is a significant
enhancement by 59.03 ng m$^{-3}$ seen during 2005 – 2013 due to a series of moderate and large
volcanoes. This figure suggests that volcanic sulfate aerosols persist for 2 – 3 years in the
stratosphere (100 – 5 hPa) after the eruption



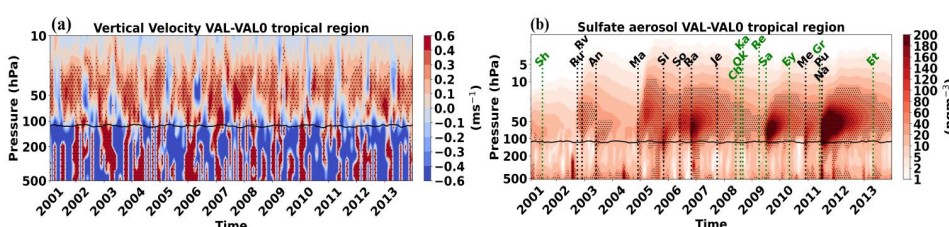

Figure 6: Time series of vertical velocity (m s$^{-1}$) anomaly (VAL - VAL0) zonally averaged over the tropical region (20° S – 20° N), (b) same as (a) but for volcanic sulfate aerosol (ng m$^{-3}$). The black dots indicate significance at 95% confidence level, obtained using a student t-test for the mean difference. Dotted vertical lines show the month of eruption. A thick black line in Figs. a-b indicates the tropopause.

The increase in stratospheric sulfate aerosol concentrations during the initial two months following the eruption in the tropics is presented in Table S1. There is an enhancement in stratospheric sulfate aerosols in the tropics due to extratropical volcanoes (table S1, S2). This may be due to transport from extra-tropics to the tropics. It should be noted that aerosols are transported from the extratropics to the tropics and vice versa (Günther et al., 2018; Timmreck, 2012; Haywood et al., 2010; Robock, 2000). However, such analysis is out of the scope of the manuscript.

**3.3 Impact of volcanic sulfate aerosol on the QBO**

In this section, we show the impact of volcanic sulfate aerosol on the QBO. Figure 7a shows vertical variation of the phases of QBO in zonal winds for the 5° S – 5° N, derived from the VAL0 simulation. It shows that downward propagation of the easterly phase of the QBO is faster than (~22 to 26 months cycle) the westerly phase of the QBO (~24 to 36 months cycle). Volcanoes cause prolongation in the easterly phase by ~12 to 20 months and the westerly phase by ~16 to 24 months, as seen from the anomalies (VAL - VAL0) plotted in Figure 7b. Figure 7b also shows the disruption of the downward propagation of QBO phases after 2012. This



may be due to less negative radiative forcing and heating rates anomalies (VAL-VAL0) (Figure
5). The changes in zonal wind phases due to volcanic aerosols (VAL - VAL0) at 32 hPa show
prolongation of easterly and westerly by ~12 to 24 months (Fig. 7c) and at 10 hPa by ~12 to
20 months (Fig. 7d).

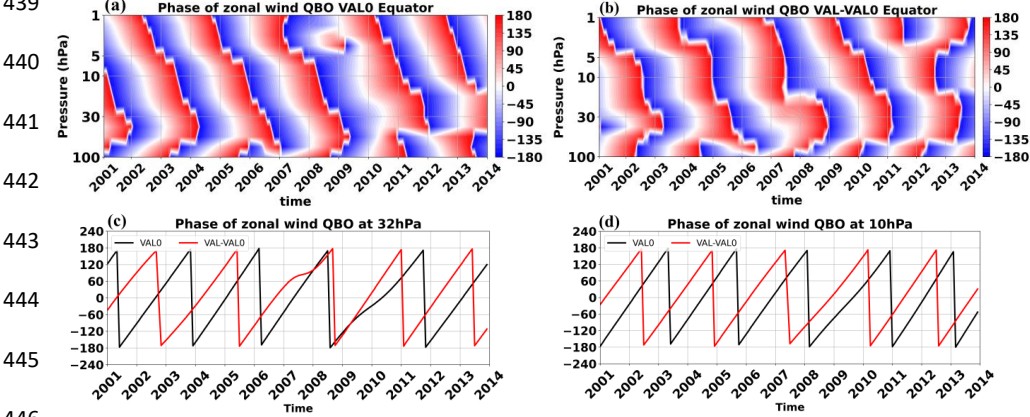

Figure 7: Time series of zonal wind phase (degree), (a) zonal wind phase form VAL0
simulation and averaged over the equatorial region (5° S – 5° N), (b) same as (a) but averaged
form VAL - VAL0 simulation, in Figs. a–b, red shading indicates the westerly phase, and blue
shading indicates the easterly phase (c) zonal wind phase at 32 hPa, (d) same as (c) but for 10
hPa.

Previous studies have also reported that tropical volcanic eruptions can delay the
progression of the QBO phases. For example, from the UM-UKCA aerosol-climate model,
Brown et al. (2023) reported tropical eruptions (e.g., Mt. Tambora in 1815 and Mt. Pinatubo in
1991) delayed the progression of the QBO phases by ~10 months.
Furthermore, we show the QBO amplitude in the zonal wind for the VAL0 simulation
(Figure 8a). The QBO amplitude shows a double-peak structure near the equator (10°S–10°N),
with maxima centered at 15 hPa (0.6 m s⁻¹) and 1 hPa (1.07 m s⁻¹).  Previous studies reported
that the double-peak structure in the zonal wind at the equator is produced by the secondary




meridional circulation induced by the QBO (Plumb and Bell, 1982; Punge et al., 2009;
Fadnavis and Beig, 2009). These studies explain the mechanism of formation of double peak
structure at the equator as: the zonal winds are assumed to be sinusoidal in the vertical direction.


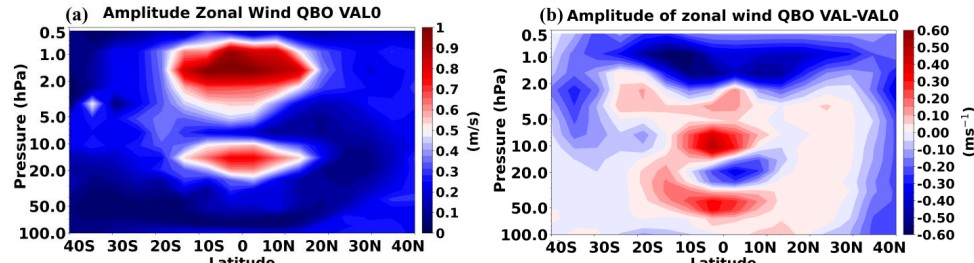

Figure 8: (a) Latitude pressure cross-section of the amplitude of QBO in zonal wind (m s$^{-1}$)
averaged for the period 2001 – 2013 from VAL0 simulation, (b) same as (a) but for VAL –
VAL0.
To maintain the thermal wind relationship, there is a warm anomaly in the westerly shear
zone and a cold anomaly in the easterly shear zone. Thus, there must exist a region of relative
sinking and rising motion in the westerly and easterly shear zones, to maintain temperature
anomalies. In the ideal condition of zonal winds, the maximum sinking/rising motion appears
in the maximum westerly/easterly shear zone.  The maximum shear zone coincides with the
zero zonal wind level. Also, the location of the maximum easterly/westerly wind coincides
with that of the maximum horizontal divergence/convergence. The sum of the QBO-induced
vertical circulation and the background annually varying extra tropically-driven residual
circulation is total residual circulation. Thus two peaks in amplitude of the QBO are expected
near maximum horizontal divergence, where the vertical velocity of the total residual
circulation is weaker at the equator than in the subtropics.
Figure 8 b shows the QBO amplitude from the VAL–VAL0 simulations. It shows that heating
induced by the volcanic sulphate aerosols affect the double peak structure in the amplitude of





the QBO. The two peaks seen in VAL0 simulations also seen anomalies (Fig. 8) but peaks are
at higher atmospheric levels 10 hPa (0.25 m s⁻¹), and 50 hPa (0.20 m s⁻¹). It is seen that the
maximum at 1 hPa near the equator in the VAL0 simulation is replaced with minimum in VAL-
VAL0. In general Figure 8 a-b shows that heating induced by volcanic sulphate aerosols alters
the phase and amplitude of the QBO near the equator.

**4. Conclusion**

The ECHAM6-HAMMOZ chemistry-climate model simulation for eruptions of

volcanoes during the period from 2001 – 2013 was analyzed against control simulations to
study the impact of volcanic sulfate aerosols on the radiative forcing and QBO.
1. Our simulations show that volcanoes during the study period elevated tropical SAOD by

0.00039 to 0.0097, leading to a negative radiative forcing in the tropical region (TOA: -

0.92±0.34 W m⁻²; surface: -0.88±0.18 W m⁻²). In agreement with our findings, other

modeling studies also demonstrate an enhancement in SAOD and a reduction in radiative

forcing. For example, Brühl et al. (2015) reported an increase in SAOD by 0.001 to 0.01

in the tropics for tropical volcanic eruptions from 2002 to 2012 using EMAC simulations,

producing a decrease in radiative forcing by -0.1 to -0.26 W m⁻². Schmidt et al. (2018)

estimated enhancement in SAOD by 0.0076 during 2005 – 2014 that caused a reduction

in global mean radiative forcing by -0.10 W m⁻² using the CESM1-WACCM model.

2. The volcanic eruptions increased the sulfate aerosol amount in the lower stratosphere by

46.95 ng m⁻³ during 2001 – 2013. These aerosols enter the tropical stratosphere and elevate

the lower stratospheric heating by 0.13±0.05×10⁻² K d⁻¹ and vertical velocity by 0.42 m

s⁻¹. The lower stratospheric heating further aids the transport of aerosols into the deep

stratosphere.



3. QBO estimated from the model simulations using the wavelet analysis shows that stratospheric heating induced by volcanic sulfate aerosols causes disruptions of the QBO phases and decreases the amplitude of the zonal wind. The easterly phase of the QBO is prolonged by ~12 to 20 months, and the westerly phase by ~16 to 24 months.

4. The secondary meridional circulation induced by the QBO produces a double-peak structure in the amplitude at the equator with peaks at 10 hPa and 50 hPa.

5. Our study points out that moderate and large volcano modules QBO. QBO also modulates tropical convection and cyclones, producing an impact on extreme weather (Fadnavis et al., 2011). Hence, it is important to include volcanic emissions and the QBO in the weather prediction model for a better forecast. Ice-core and satellite measurements suggest that future explosive volcanic eruptions could inject more sulfur dioxide into the stratosphere over 2015 – 2100 than current standard climate projections (i.e., ScenarioMIP) (Chim et al., 2023). The injection of more sulfur dioxide into the stratosphere will cause large impacts on tropospheric weather.

**Authors' contribution:** S.F. and A. L. designed the modelling experiments. P.C. analysed the model output. All authors contributed to the writing of the manuscript.

**Conflict of Interest:** More than one author is Editor of the journal Atmospheric Chemistry and physics.

**Data Availability Statement:** The data presented in this study can be obtained from the corresponding author upon request. The SAOD data by Global Space-based Stratospheric Aerosol Climatology (GloSSAC) can be downloaded from https://asdc.larc.nasa.gov/project/GloSSAC. The zonal wind data by ERA5 can be downloaded from https://cds.climate.copernicus.eu/datasets/reanalysis-era5-pressure-levels-monthly-means?tab=download.



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
