# Peer review of "Impact of volcanic sulfate aerosols on the stratospheric heating: implications on the 1 2 Quasi-Biennial Oscillation 3 4 Prashant Chavan1,2, Suvarna Fadnavis1\*, Anton Laakso3, Jean-Paul Vernier4,5, 5 Simone Tilmes6, and Rolf Müller7</s"

_EGUsphere, 2024_

## Referee Comment (RC1)

Review

The concept of this study is very interesting but as it stands, I am not convinced of the effect of recent volcanic activity on the QBO. The difference between the QBO in simulations with and without volcanic eruptions seems mostly to happen at the onset of the simulation, rather than in response to large volcanic activity happening after 2005. In my opinion, more analysis is needed to understand what is happening in the simulations and why. Here, I don't find the conclusions sufficiently supported by the results. Furthermore, the final section of the study (Fig. 8) is a topic I am very interested in, but it is not clear what the findings of this work are or if any new conclusions have been reached. I think to meaningfully add to existing understanding, more than a simple lat-height anomaly is required. Beyond this, some statements would benefit from additional citations and the supplementary could easily be extended to include the analysis suggested above.

With these additions, I would find the study fits in the scope of the journal, and is well-structured with a sufficient description of methods.

Major comments:

I remain a bit confused with the statement 'QBO is not included' in the ECHAM model and whether that is an issue. Why did you choose to use this model over a model that has an internally generated QBO? What are the possible issues with not having the QBO and deriving it from wavelet analysis?

Could you compare to observations to confirm your model results? The present day QBO reanalysis and observations are available.

I am unconvinced of the impact of the eruptions on the QBO and vertical winds. Why do larger eruptions not seem to have a larger effect on vertical velocity or the QBO? In addition to the vertical velocity anomaly, I would be interested to see the temperature anomaly and perhaps even the latitudinal temperature gradient.

The wavelet analysis and how it gets a QBO from the zonal wind is difficult for me to follow. Some of the mathematical terms are not defined and the purpose of each equation is not clearly explained.

Fig. 6: There is almost no difference between sulfate burden in VAL and VAL0 between 2001 and 2005, however the vertical velocity increases right from the start of the simulation and doesn't seem to increase any further after 2005 or after larger eruptions. The relationship between sulfate and vertical velocity is not that convincing to me.

Fig. 7 is not intuitive to understand. It may benefit from showing VAL, VAL0 and the anomaly, since the anomaly is hard to interpret on its own. Panels c and d are also hard to understand. In Panel d, the anomaly and VAL0 look identical with a small shift, suggesting VAL0 is slightly out of sync with VAL from the start but not necessarily suggesting an increased QBO period over the course of the simulation. To indicate an impact of volcanic eruptions, I would expect the anomaly to have a different period to VAL0 and maybe even to change over time in response to difference eruptions. I am happy to be proven wrong, but I think more work needs to be done to confirm the impact of the eruptions.

The section describing Fig. 8 needs some work. What are the consequences of your finding that the zonal mean cross section changes? Is it important? Is it significant? Why explain the thermal-wind balance principles but not discuss or show figures of temperature or wind anomalies?

The thermal wind relationship and the height-latitude structure of the QBO is related to the meridional temperature gradient. To explain the changes in Fig. 8, I would guess the sulfate aerosol has caused a change in the temperature gradient (Brown et al., 2023 has a detailed explanation). In general, I find the lat-height changes to the QBO of great interest but I am not sure of what message you are trying to convey here.

Minor comments

- L221: Why do you do the wavelet analysis on the anomaly? Surely this is not certain to have an amplitude or period.
- The model evaluation section 3.1 needs a slightly more nuanced analysis in my opinion. The baseline SAOD is underestimated by the model but it simulates increases in response to volcanic eruptions that are much larger than observed. This could lead to overestimation of the volcanic effects. Adding a 1:1 line to Fig. b would demonstrate this.
- L286: Why would output on different levels create a bias?
- L288: How do you know transport and resolution processes can produce the difference in your model vs GloSSAC?
- L298-306: Why do you just compare the year 2008? To me this is not the best way to decide if the QBO can be well represented. I would think the higher latitude zonal winds are not so important and may even compare a zonal mean zonal wind at all pressure levels, similar to Fig. 8.
- L432: A disruption of 12 months between which time period? Or every cycle of the QBO is extended?

- L434: Is this an observed phenomenon or just in your simulation?
- L492: sentence doesn't read well
- L493: higher atmospheric pressure but lower atmospheric level
- L497-489: How does Fig. 8 add to this background knowledge? Is this paragraph relevant or is it better suited to the introduction?

Conclusion section:

- I think point 1 is fine as it is, but to be a great point perhaps you could add what you have done that is different to existing results on this topic.
- Point 3, similar to an earlier comment: is 12 – 24 months an accumulated delay over the course of the simulation? There isn't any evidence of responses from specific eruptions in this work.
- Point 4 is not a new finding of your study as far as I am aware. It is just a description of the zonal mean zonal wind at the equator when you average over a long period of time.
- For point 5, is 'large' impacts on weather perhaps an overstatement.

---

## Author Comment (AC1)

Fig.: (a) Time series of temperature zonally averaged over the 5S-5N for the period 2001-2013. (b) Latitude-pressure cross section of zonally averaged temperature for period 2001-2013 from VAL-VAL0 simulation. The black dots indicate the regions where the differences are significant for the difference of the mean (p-value < 0.05) using a two-sided t-test.